# Entanglement and manipulation of the magnetic and spin–orbit order in multiferroic Rashba semiconductors

J. Krempaský[1], S. Muff[1,2], F. Bisti[1], M. Fanciulli[1,2], H. Volfová[3], A.P. Weber[1,2], N. Pilet[1], P. Warnicke[1], H. Ebert[3], J Braun[3], F. Bertran[4], V.V. Volobuev[5,7], J. Minár[3,6], G. Springholz[7,*], J.H. Dil[1,2,*] & V.N. Strocov[1,*]

Entanglement of the spin–orbit and magnetic order in multiferroic materials bears a strong potential for engineering novel electronic and spintronic devices. Here, we explore the electron and spin structure of ferroelectric α-GeTe thin films doped with ferromagnetic Mn impurities to achieve its multiferroic functionality. We use bulk-sensitive soft-X-ray angle-resolved photoemission spectroscopy (SX-ARPES) to follow hybridization of the GeTe valence band with the Mn dopants. We observe a gradual opening of the Zeeman gap in the bulk Rashba bands around the Dirac point with increase of the Mn concentration, indicative of the ferromagnetic order, at persistent Rashba splitting. Furthermore, subtle details regarding the spin–orbit and magnetic order entanglement are deduced from spin-resolved ARPES measurements. We identify antiparallel orientation of the ferroelectric and ferromagnetic polarization, and altering of the Rashba-type spin helicity by magnetic switching. Our experimental results are supported by first-principles calculations of the electron and spin structure.

[1] Swiss Light Source, Paul Scherrer Institut, CH-5232 Villigen PSI, Switzerland. [2] Institute of Physics, École Polytechnique Fédérale de Lausanne, CH-1015 Lausanne, Switzerland. [3] Department of Chemistry, Ludwig Maximillian University, 81377 Munich, Germany. [4] SOLEIL Synchrotron, L'Orme des Merisiers, F-91192 Gif-sur-Yvette, France. [5] National Technical University, Kharkiv Polytechnic Institute, Frunze Str. 21, 61002 Kharkiv, Ukraine. [6] New Technologies-Research Center University of West Bohemia, Plzeň, Czech Republic. [7] Institut für Halbleiter-und Festkörperphysik, Johannes Kepler Universität, A-4040 Linz, Austria. * These authors jointly supervised this work. Correspondence and requests for materials should be addressed to J.K. (email: juraj.krempasky@psi.ch).

The interplay between electronic eigenstates, spin and orbital degrees of freedom, combined with fundamental breaking of symmetries is currently one of the most exciting fields of research. It not only forms the basis for giant magnetoresistance[1], spin-torque manipulation of magnetic domains[2] and the use of Rashba effects for spin manipulation[3], but has also lead to outstanding recent discoveries of new quantum phases such as topological insulators[4], Weyl semimetals[5] and Majorana fermions[6]. While already each of these topics has initiated new research fields, the combination of several of these effects in a single material opens up a new realm of opportunities. For example, superconductors with sizable Rashba spin splitting and ferromagnetic (FM) order would unite all ingredients for formation and manipulation of so-called anyons, generalizing the concept of Majorana fermions[7]. Multiferroics, such as $Ge_{1-x}Mn_xTe$ (ref. 8), fulfil these requirements providing unusual physical properties[9–11] due to the coexistence and coupling between FM and ferroelectric (FE) order in one and the same system. Here we show that multiferroic $Ge_{1-x}Mn_xTe$ inherits from its parent FE $\alpha$-GeTe compound a giant Rashba splitting of three-dimensional (3D) bulk states[12–14], which competes with the Zeeman spin splitting induced by the magnetic exchange interactions. The collinear alignment of FE and FM polarization leads to an opening of a tunable Zeeman gap of up to 100 meV around the Dirac point of the Rashba bands, coupled with a change in spin texture by entanglement of magnetic and spin–orbit order. Through applications of magnetic fields, we demonstrate manipulation of spin-texture by spin-resolved photoemission experiments, which is also expected for electric fields based on the multiferroic coupling. The control of spin helicity of the bands and its locking to FM and FE order opens fascinating new avenues for highly multifunctional multiferroic Rashba devices suited for reprogrammable logic and/or non-volatile memory applications.

Similar to how the magnetic moment of an unpaired electron strongly enhances the exchange interaction, the local spin–orbit interaction is greatly enhanced by broken inversion symmetry. The larger the symmetry breaking, the larger the Rashba spin splitting becomes[15–17]. This is exemplified by the giant spin splitting observed for the polar semiconductor BiTeI (refs 18,19), the polar surface of $SrTiO_3$ (ref. 20), as well as for high-Z surface alloys[21,22]. In FE materials, the inversion symmetry is naturally broken by the FE polarization produced by the relative anion/cation displacements in the crystal lattice. Thus, the spin splitting in ferroelectric Rashba semiconductors (FERS) can assume exceedingly high values. $\alpha$-GeTe exhibits record splitting values[12–14,23] due to the very large rhombohedral lattice distortion providing cation/anion displacements as large as 10% of the (111) lattice plane spacing (Fig. 1a). The spin splitting of its bulk bands is intimately linked to the FE polarization, meaning that it can be switched and controlled by reversing the FE polarization direction[12]. Doping of GeTe with Mn maintains the rhombohedral lattice distortion and FE in $Ge_{1-x}Mn_xTe$ up to Mn concentrations as high as 30% (ref. 8). At the same time, the Mn spins couple to each other via free carrier mediated RKKY (Ruderman-Kittel-Kasuya-Yosida) exchange interactions. As a result, FM ordering as in (GaMn)As (ref. 24) occurs, which turns $Ge_{1-x}Mn_xTe$ into a multiferroic Rashba semiconductor (MUFERS) that combines both FE and FM properties as shown by Fig. 1e,f, respectively. In fact, multiferroic $Ge_{1-x}Mn_xTe$ features one of the highest FM Curie temperatures[8,25] and highest reported magnetic moments amongst diluted ferromagnetic semiconductors (FMS)[26].

FM ordering and FE symmetry breaking both lead to a spin splitting of the electronic band structure. For FMS, the spontaneous magnetization leads to a vertical Zeeman splitting of the bands (Fig. 1b), whereas for FERS the electric polarization

induces a Rashba splitting in the $k_{\parallel}$ direction (Fig. 1c). In MUFERS both effects are entangled with each other, modifying the spin texture and band dispersions (Fig. 1d). For a basic discussion, we consider a simplified Rashba–Zeeman (RZ) model based on a two-dimensional (2D) free-electron approximation (RZ gas) in which the energy eigenvalues $E_{\pm}(k)$ of the Rashba split bands in the presence of magnetic order are given by

$$E_{\pm}(k) = E_0 + \frac{\hbar^2 k^2}{2m^*} \pm \sqrt{\Delta_Z^2 + 4\alpha_R^2 k^2}, \qquad (1)$$

where $E_0$ is the energy of the band edge, $m^*$ the effective electron mass, $\alpha_R$ the $k$-linear Rashba coupling constant and $\Delta_Z$ the momentum-independent Zeeman gap near $\bar{\Gamma}$. In principle, any orientation of the magnetization opens a gap near $\bar{\Gamma}$ (Fig. 1d). Considering the momentum-dependent expected values of the spin components in all directions, however, only systems with magnetization parallel to the inversion symmetry breaking direction can reorient the spin of the Rashba electrons and open a gap in all $k_{\parallel}$ directions as illustrated by Fig. 1g and Supplementary Fig. 1 (Supplementary Note 1.1). Due to coupling between the easy axis of magnetization and the rhombohedral lattice distortion, (111) $Ge_{1-x}Mn_xTe$ films fulfil these criteria[8] because the electric polarization and magnetic moments are colinear and perpendicular to the surface (Fig. 1e,f). Since the helicity of the Rashba effect is locked to the FE moments[13,14,27], multiferroic switching is expected to entangle with the spin helicity as sketched in Fig. 1g,h.

## Results

**Soft-X-ray ARPES.** For a systematic spectroscopic study of the RZ effect in this multiferroic system, we have prepared epitaxial $Ge_{1-x}Mn_xTe$ films with various Mn content up to 13% ($x_{Mn} = 0.03, 0.054, 0.08$ and $0.13$, see 'Methods' section). We use SX-ARPES to elucidate the local electronic structure of the Mn ions and RZ splitting with 3D $\mathbf{k}$-space resolution, and spin-resolved ARPES (SARPES) in the ultraviolet photon energy range to explore the corresponding changes in the spin textures in this multiferroic material (see 'Methods' section). Figure 2 summarizes our SX-ARPES data obtained with photon energies from 340 to 800 eV. A fundamental advantage of this energy range[28] is an increase of the photoelectron escape depth and thus of the intrinsic definition of the ARPES experiments in surface-perpendicular momentum $k_{\perp}$ (ref. 29) crucial for observation of the inherently 3D electronic structure of $Ge_{1-x}Mn_xTe$. This is illustrated in Fig. 2a by showing the dispersive spectral weight at the Fermi level ($E_F$) and at a binding energy of 700 meV as a function of $k_{\perp}$ varied through photon energy. This map, that is similar for all samples except for an increase of the spectral broadening with increasing $x$, readily identifies the Z-points in the 3D Brillouin zone where the Rashba splitting is most pronounced[14]. Measured in the Z-point at $h\nu = 480$ eV, where the spectral intensity is maximal, the ARPES spectral weight along the A–Z–A direction is shown in Fig. 2b. This map clearly shows the pair of Rashba split bands in the vicinity of the Z-point consistent with the theoretical expectation (Fig. 2c). Another virtue of SX-ARPES is its chemical specificity achieved with resonant photoemission. The corresponding data measured at the Mn L-edge as indicated by the X-ray absorption (Fig. 2d) demonstrates that the Mn $3d$ states hybridize into the GeTe host states through the whole valence band, in particular into the Rashba bands below the Zeeman gap (see Supplementary Fig. 3 in Supplementary Note 2). In particular, we note the absence of any impurity states in the vicinity of $E_F$ which might otherwise interfere with the RZ splitting in this region.

Our further discussions of the Zeeman gap opening is based on Fig. 2f, which presents the experimental band structure along

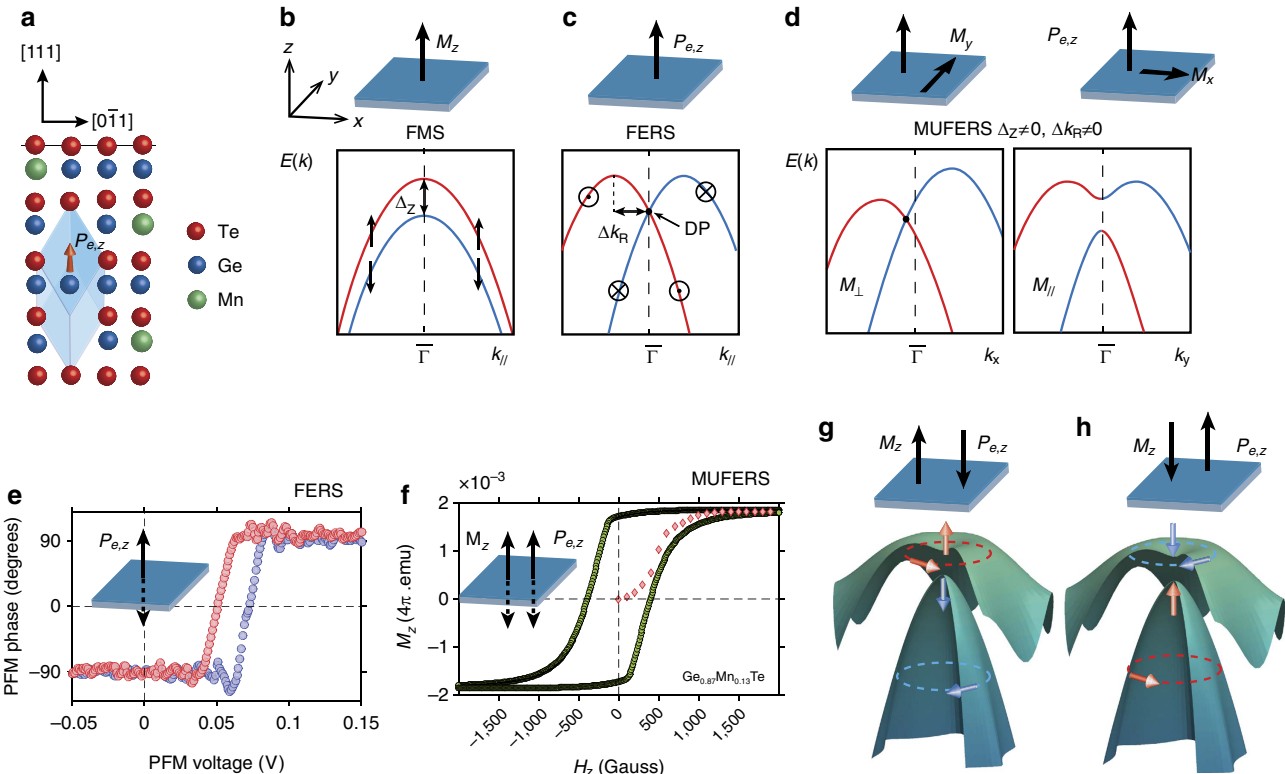

**Figure 1 | Basic properties.** (**a**) Sketch of multiferroic $Ge_{1-x}Mn_xTe$ with FE displacement of Ge(Mn)-atoms inside the rhombohedrally distorted unit cell along [111] as indicated by the orange arrow. (**b**) Schematic Rashba-gas band maps of a ferromagnetic semiconductor (FMS) compared with a ferromagnetic Rashba semiconductor (FERS) (**c**) and a multiferroic Rashba semiconductor (MUFERS) in **d**, with their dependence on the orientations of the FM ($M$) and FE ($P_e$) order (see text). (**e**) Typical out-of-plane FE phase hysteresis measured by piezo-force-microscopy for FE GeTe and preserved in $Ge_{1-x}Mn_xTe$ (ref. 8). (**f**) Out-of-plane FM hysteresis curve of multiferroic $Ge_{0.87}Mn_{0.13}Te$ measured by SQUID. (**g,h**) Band plots of the two upper occupied RZ-split valence bands as a function of **k**-vectors around Z-point of $Ge_{1-x}Mn_xTe$ and corresponding spin-texture switching upon colinear reversal of $M_z$ and $P_z$.

A–Z–A in vicinity of the Z-point, measured through the whole sample series. To elucidate the correlation between Rashba splitting and Zeeman gap, and its dependence on Mn doping, we compare in Fig. 3 the zoom-in band maps around the Z point near $E_F$ (dashed rectangle in Fig. 2b) with *ab initio* calculations. As detailed in Supplementary Fig. 4, careful experimental alignment is needed for the quantitative assessment of the Zeeman gap. To derive the competing Rashba and Zeeman terms $\alpha_R$ and $\Delta_Z$ for different Mn concentrations, both theoretical and experimental data in Fig. 3 are fitted with the simplified RZ gas model described above (red–blue lines). Due to the simplicity of the free electron model that neglects both third order corrections in momentum, and the strong band non-parabolicity of the IV–VI compounds, the fit does not reproduce the actual dispersions but it is accurate for small momenta and yields direct values for both $\alpha_R$ ($\pm 7\%$) and $\Delta_Z$ ($\pm 10$ meV), listed in Supplementary Table 1. More accurate fits based on a massive Dirac fermion seen in light-blue are listed in Supplementary Table 2 and discussed in Supplementary Note 1.2. As fitted results from both models are consistent (Supplementary Fig. 2), we discuss next the Mn-concentration dependence of the $\alpha_R$ and $\Delta_Z$ based on the simplified model.

We observe both in experiment and theory that a Zeeman gap, absent in $\alpha$-GeTe, appears and widens with increasing Mn concentration, that is, increasing film magnetization. This is clear-cut evidence that the Zeeman gap opening of the Rashba bands is induced by the FM order of the multiferroic system. To the best of our knowledge, this is the first experimental confirmation of the opening of a Zeeman gap at the Dirac point in a bulk system with strong FM order. In magnetically doped

topological insulators such results are still being debated due to the lack of measurements justifying the FM ordering of the dilute dopants at the ARPES measurement conditions[30–32]. Plotting in the lower right panels of Fig. 3a,b, the resulting values for $\alpha_R$ and $\Delta_Z$ as a function of Mn content reveals that the Zeeman gap reaches remarkably high values around 100 meV for $x_{Mn} > 10\%$. Furthermore, the progression of the Zeeman gap is accompanied by a decrease of the Rashba splitting in both fitting models. The microscopic origin of this trend is that higher Mn doping reduces the rhombohedral distortion of the $Ge_{1-x}Mn_xTe$ lattice[8]. This, in turn, reduces the FE polarization (cf. Fig. 1a) and thus, the strength of the Rashba coupling constant. For the host $\alpha$-GeTe(111) we find that $\alpha_R$ is as high as 4.3 eVÅ, which is in excellent agreement with theoretical predictions by Picozzi[23]. We also note that since the experimental Zeeman gap appears to saturate for $x_{Mn}$ around 10%, our conjecture is that higher Mn doping might lead to Mn-phase segregation in the undoped host GeTe lattice[25,33] and that at high $x_{Mn}$ antiferromagnetic coupling between neighbouring Mn atoms reduces the average FM moment per Mn atom[34].

A particularly interesting feature for device applications is the fact that with increasing Mn doping the Zeeman gap shifts towards $E_F$. This will reduce the bias voltage needed to move $E_F$ inside this gap and create a single spin polarized Fermi surface reminiscent of an ideal topological insulator[4]. In contrast to the 2D surface state of a topological insulator, however, $Ge_{1-x}Mn_xTe$ has a 3D Fermi surface, which in the absence of FM order takes the form of a spin polarized spindle torus[19,35]. The constant energy surface shows a strong hexagonal warping which becomes three-fold away from the $\Gamma$ and Z points, in

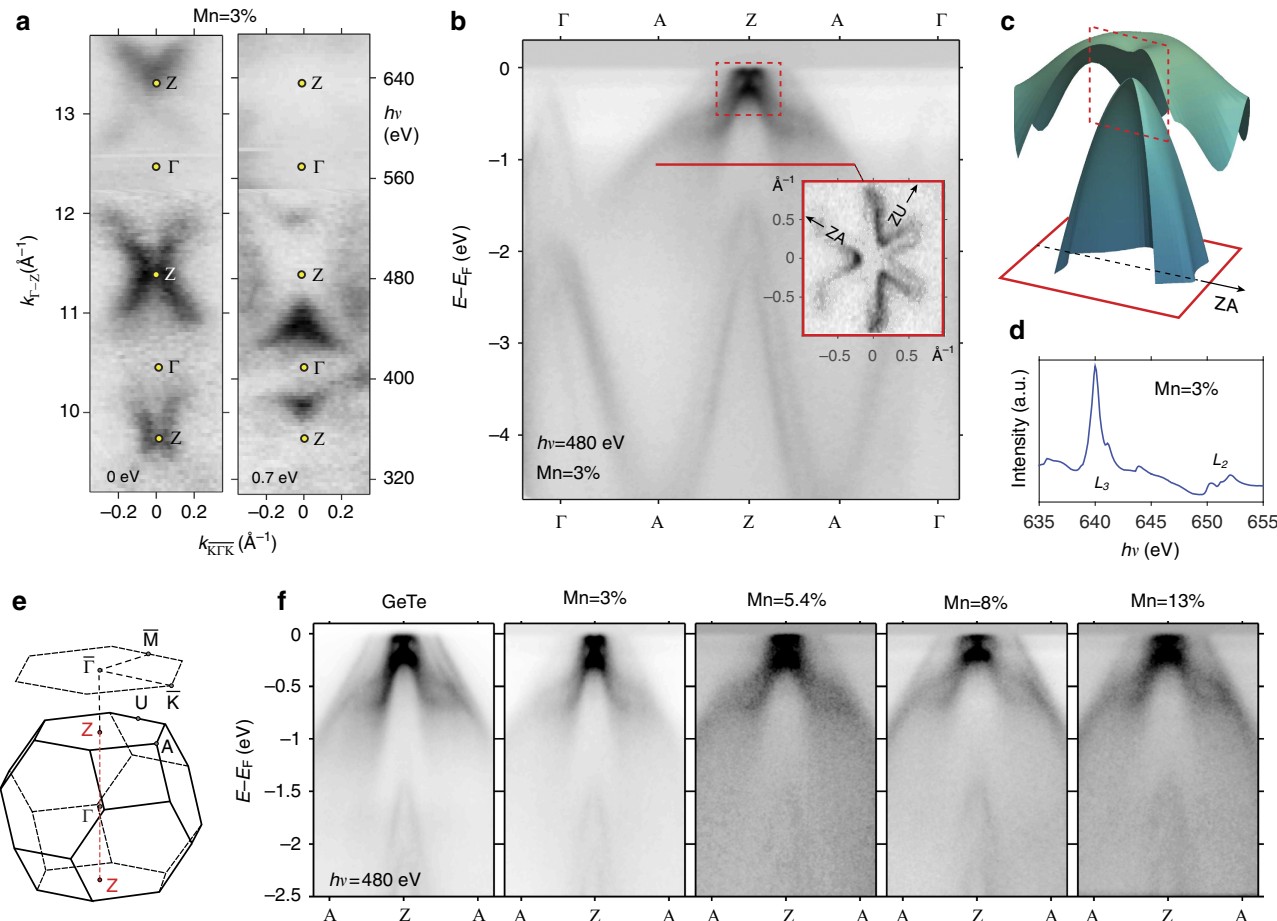

**Figure 2 | Soft X-ray ARPES data.** (**a**) SX-ARPES photon-energy dependent constant binding-energy maps of the $Ge_{0.97}Mn_{0.03}Te$ valence band bulk states near Fermi level and at 0.7 eV binding energy in the $\overline{K\Gamma K}$ plane. (**b**) SX-ARPES band map measured along Z–A–$\Gamma$ with theoretical band plots in **c** to compare with measured data near $E_F$. The inset is a constant binding energy map at 1 eV binding energy. (**d**) X-ray absorption data near the $L_3$ absorption edge. (**e**) Brillouin zone of quasicubic GeTe. (**f**) ARPES band maps along A–Z–A for the selected Mn dopings. The dashed rectangle in **b** zooms into region near $E_F$ where the Zeeman gap opening is examined.

accordance with the crystal symmetry. In Fig. 3c the 3D Fermi surfaces of $Ge_{1-x}Mn_xTe$ are shown for three different chemical potentials, taking into account both the deviation from an ideal torus due to the warping and the opening of a Zeeman gap. For $\mu > \frac{1}{2}\Delta_Z$ the gap opening does not influence the Fermi surface, and for $\mu < -\frac{1}{2}\Delta_Z$ the gap is only visible for a limited $k_\perp$ range because of the dispersion to higher binding energy away from Z. For $\mu = 0$, however, the chemical potential is within the Zeeman gap and the 3D Fermi surface obtains a non-trivial spin texture due to the presence of only one single non-degenerate Fermi sheet. The exact consequences of this configuration require further study, but in both one-dimensional and 2D equivalents this leads to a template for the formation of Majorana fermions.

**Spin-resolved ARPES measurements**. We now turn to spin-resolved information achieved with ultraviolet-SARPES to show that $Ge_{1-x}Mn_xTe$ still retains the Rashba-type spin splitting from the host GeTe (Supplementary Fig. 5), but exhibits an out-of-plane spin reorientation due to FM ordering. According to previous studies, α-GeTe already exhibits a complex surface electronic structure where surface states coexist with a variety of bulk-derived states[13,14]. For $Ge_{1-x}Mn_xTe$, the data measuring the in-plane spin polarization shown in Supplementary Fig. 6 indicate opposite spin helicity of the

bulk states above and below the Zeeman gap as sketched Fig. 1g. Thus, the in-plane spin polarization is similar as for α-GeTe above and below the Dirac point[14].

To resolve the question of how the presence of FM order affects the $Ge_{1-x}Mn_xTe$ spin texture, Fig. 4a presents $P_z$ calculations for four different Mn concentrations based on the one-step photoemission approach (see 'Methods' section). Clearly, FM ordering leads to a progressive $P_z$ spin reorientation with increasing Mn concentration. In particular, the characteristic $P_z$ hexagonal warping typical for GeTe is smeared out at $\overline{\Gamma}$ (Supplementary Fig. 7). The theoretical and experimental $P_z$ spinors are visualized in Fig. 4c,d as spin-resolved energy distribution curves measured across the Zeeman gap at normal emission. Apart from difference in amplitude due to inelastic scattering effects, our experimental data and the theoretical results represented by Fig. 4c inset consistently reproduce the characteristic $P_z$ spin reorientation wiggle across the Zeeman gap induced by spontaneous FM order.

The virtue of MUFERS obviously lies in the ample possibilities for spin manipulation using electrical as well as magnetic fields or by combination of both (Fig. 1). The latter is demonstrated by Fig. 4d, where we present and compare the $P_z$ spin texture measured before and after magnetization reversal, that is, after switching the magnetization between the [111] and [$\overline{1}11$] directions with magnetic fields $\pm B$ applied perpendicular to

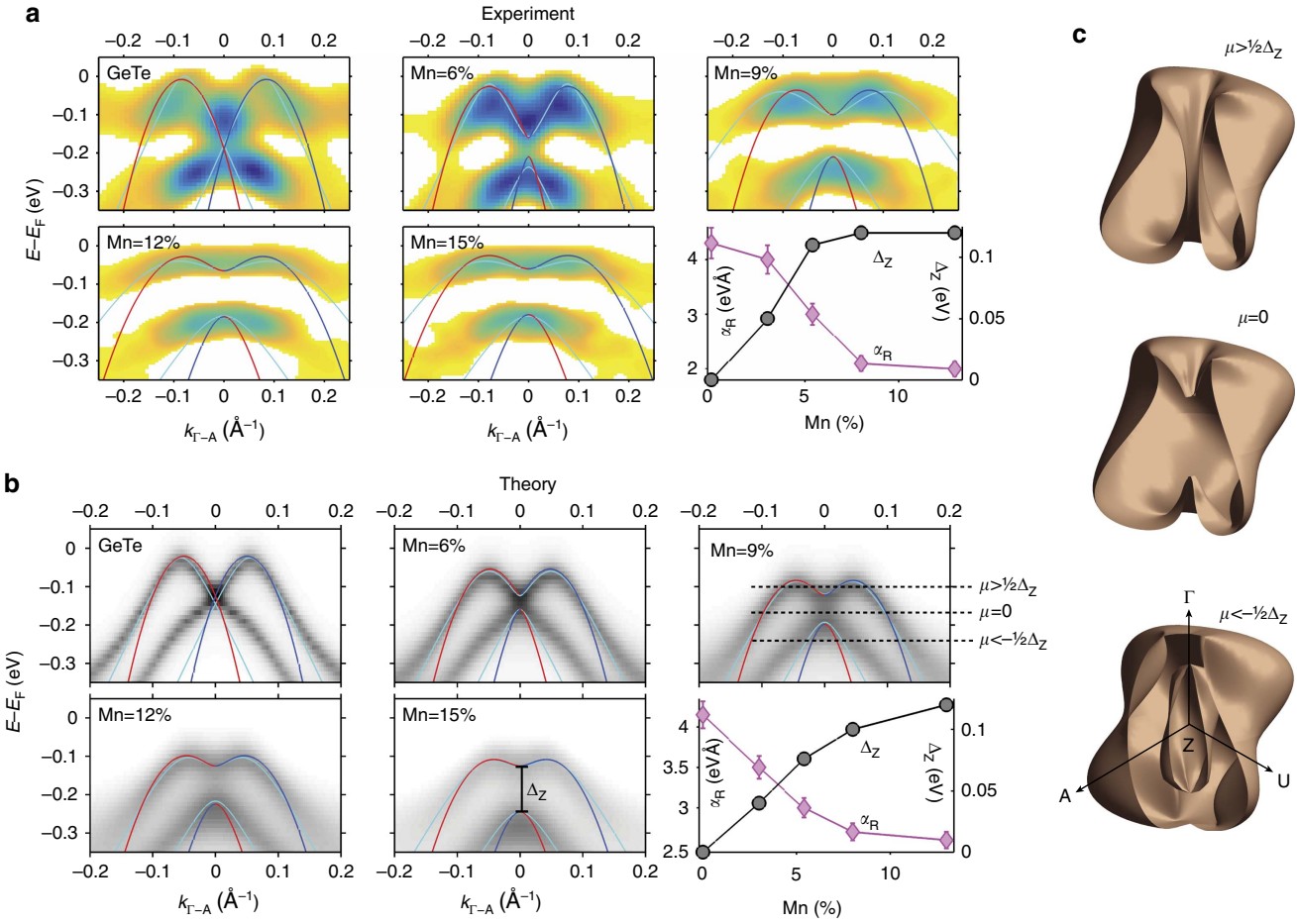

**Figure 3 | Rashba–Zeeman (RZ) splitting: experiment, theory and consequences for 3D Fermi surface.** (**a**) Zoomed-in ARPES data from Fig. 2f (second-derivative plots from red rectangle area in Fig. 2b) compared with *ab initio* $Ge_{1-x}Mn_xTe$ calculations for selected Mn dopings in **b**; each fitted to a simplified RZ gas model (red/blue lines) and more accurate Dirac fermion model (light-blue lines). The best-fit parameters of the Rashba parameter $\alpha_R$ and the Zeeman gap $\Delta_Z$ are displayed in the corresponding lower right hand panels. The $\alpha_R$ error bars were obtained from varying the fit parameters. (**c**) 3D Fermi surfaces for three different chemical potentials $\mu$.

the sample. After application of $B$ along $[\bar{1}\bar{1}\bar{1}]$, the $P_z$ polarization is significantly enhanced, meaning that the intrinsic FM order is into the sample. After switching the magnetization in the opposite [111] direction, the $P_z$ spin-texture reverts, which clearly confirms the entanglement of the spin-splitting as sketched in Fig. 1g,h. To demonstrate the $P_z$ spin reorientation around $\bar{\Gamma}$, the SARPES data in Fig. 4d was measured for finite momenta indicated by the vertical dashed line in Fig. 4b. The data shows exactly the same $P_z$ modulation as in normal emission, which is in sharp contrast to pure $\alpha$-GeTe where the out-of-plane spin texture is dominated by warping and photoemission effects[14] (Supplementary Fig. 7). According to Fig. 1f, for $Ge_{0.87}Mn_{0.13}Te$ magnetic field strengths of less than 1,000 Gauss are sufficient for this reversal process, but the actually required switching field depends also on temperature and Mn concentration[8].

The coupling between the FM and FE order implies that also the Rashba-like spin texture is influenced by the magnetic field switching. Due to the intricate mixture of states with both bulk derived and surface character at finite momenta[14], measurements of this change are rather complex with currently available experimental techniques. However, as shown in Supplementary Fig. 8, the canted Rashba-type spin texture allows us to indirectly also verify this aspect of the MUFERS.

## Discussion

In a broader perspective, our work introduces a paradigm of MUFERS, which generalizes and expands the concept of FERS and FMS. The unveiled RZ splitting has far reaching consequences both for fundamental physics and device applications. From a theoretical point of view, the Zeeman gap of MUFERS may give rise to excitations with the characteristics of localized Majorana fermions. In terms of applications, adding magnetism to FE FERS opens additional degrees of freedom and allows to strongly enhance the fidelity of spin control through additional Larmor precession of spins injected in field effect spin transistors. Spin control in such devices can be attained through electric or magnetic means, as well as by combinations of both. This vastly enhances the functionality of MUFERS devices beyond that of FM or FE systems only[36]. Thus, our results will pave the way for a new field of MUFERS systems, bringing new multifunctional assets for spintronic device applications.

## Methods

**Sample preparation method.** Experiments were performed on 200 nm thick $Ge_{1-x}Mn_xTe$ films grown by molecular beam epitaxy on $BaF_2(111)$ substrates[8,25,37]. A protective stack of amorphous Te- and Se-capping layers with a total thickness of 20 nm was used to avoid surface oxidation and degradation. It was completely removed in the ultrahigh ARPES vacuum chamber by annealing

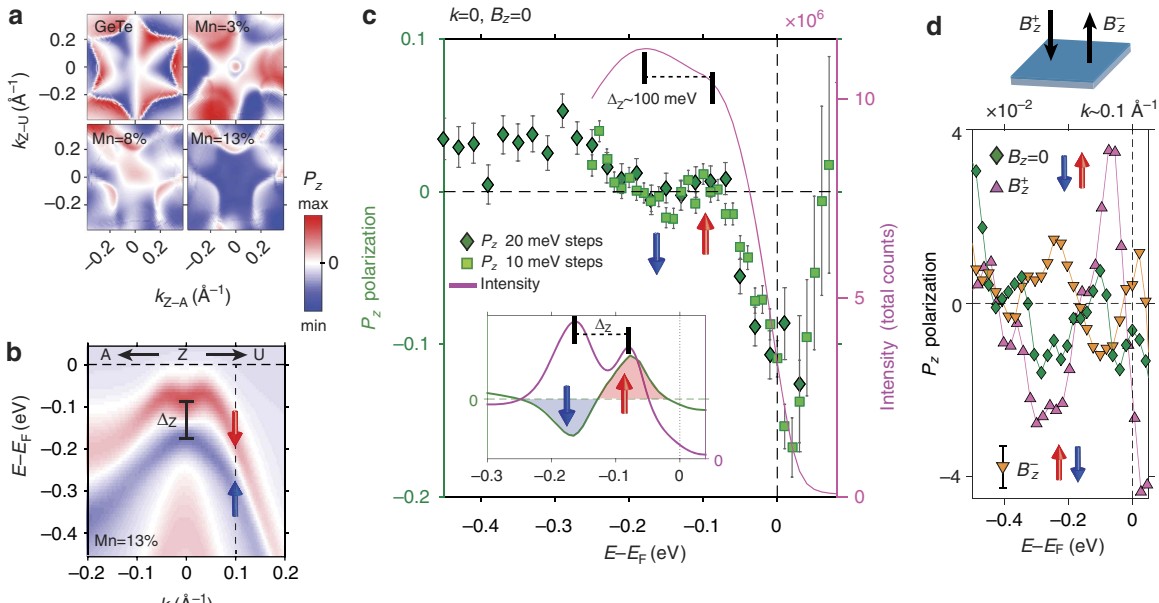

**Figure 4 | Intrinsic and magnetized out-of-plane spin polarization: theory and experiment.** (**a**) Calculated out-of-plane spin polarization in the photoemission final states for $hv = 22\,eV$ for GeTe below the Dirac point and for $Ge_{1-x}Mn_xTe$ with $x_{Mn}$ of 3, 8 and 13% below the Zeeman gap ($\approx 0.2\,eV$ binding energy). (**b**) $Ge_{0.87}Mn_{0.13}Te$ initial-state calculations based on multiple scattering alloy theory along Z–A and Z–U. (**c**) Corresponding measured SARPES $P_z$ polarization in the Z-point visualized as energy-distribution curves in $P_z$ and total intensity measured across the Zeeman gap. The theoretical data from **b** is shown for comparison in the inset. The data points correspond to two independent experimental data sets with 20 and 10 meV energy steps (the smaller step size data is a verification of the larger one in a separate measurement, error bars are obtained from measurement statistics). The vertical arrows highlight the $P_z$-wiggle resolved across the Zeeman gap $\Delta_Z$ of the order of 100 meV. (**d**) A comparison of $P_z$-spin reorientation between intrinsic and magnetized $Ge_{0.87}Mn_{0.13}Te$ along the $[\bar{1}\bar{1}\bar{1}]$ ($B_z^+$) and $[111]$ ($B_z^-$) directions.

the samples for 30–45 min at 250 °C. Comparison of SX-ARPES spectra measured through the protective Te/Se stack and on the uncapped samples evidenced that the annealing did not change the $Ge_{1-x}Mn_xTe$ bulk electronic structure but only unleashed the otherwise suppressed surface states.

**Experimental techniques.** The bulk-sensitive SX-ARPES experiments have been carried out at the SX-ARPES endstation[38] of the high-brilliance ADvanced RESonance Spectroscopies beamline at the Swiss Light Source synchrotron radiation facility, Paul Scherrer Institute, Switzerland, using $p$-polarized soft-X-ray photons in the energy range 300–800 eV. The slit of the hemispherical photoelectron analyser PHOIBOS-150 was oriented in the scattering plane including the incident photons and detected photoelectrons. The combined (beamline and analyser) energy resolution was 80 meV, and analyser angular resolution 0.07°. The experiments were carried out at low temperature of 12 K to quench the thermal effects destroying the coherent $k$-resolved spectral component at high photon energies[39]. The complementary SARPES experiments were performed at the COPHEE end-station of the Surfaces and Interfaces Spectroscopy beamline at SLS using $p$-polarized photons in the energy range 20–25 eV. The Omicron EA 125 hemispherical energy analyser was equipped with two orthogonally mounted classical Mott detectors[40]. The whole set-up allows simultaneous measurements of all three spatial components of the spin-polarization vector for each point of the band structure. The SARPES data were measured with the sample azimuths Z–U or Z–A aligned perpendicular to the scattering plane. The SARPES error bars indicated in figures are given by $1/(S\sqrt{N})$, where $S$ is the detector Sherman function (0.08 at COPHEE) and $N$ is the detector count rate. The angular and combined energy resolution were 1° and 60 meV, respectively. In spin-integrated mode these resolutions were set to 0.5° and 20 meV. SARPES data from magnetized samples were measured at Soleil CASSIOPÉE beamline where $S = 0.12$. All data were taken at 20–30 K. The experimental results in the main text were reproduced on fourteen samples with individual annealing preparation over different experimental runs. The magnetic properties were measured using a superconducting quantum interference device (SQUID) in vibrating sample magnetometry mode. Magnetic hysteresis loops were recorded at 5 K with the applied field directed perpendicular to the sample plane. The piezo-force microscopy was performed at the NanoXAS endstation at the SLS using plain platinum tip at room temperature.

**First principles calculations.** The *ab-initio* calculations are based on the multiple scattering approach (Korringa–Kohn–Rostoker (KKR) method) and density

functional theory[41]. Spin–orbit coupling has been naturally included by use of a fully relativistic four-component scheme. As a first step of our investigations we performed self-consistent calculations for 3D bulk as well as 2D semi-infinite surface of $Ge_{1-x}Mn_xTe$ within the screened Korringa–Kohn–Rostoker formalism[41]. The corresponding ground state band structures are presented in terms of Bloch spectral functions. The self-consistent results served as an input for our spectroscopic investigations. The ARPES calculations were performed in the framework of the fully relativistic one-step model of photoemission[42] in its spin-density matrix formulation, which accounts properly for the complete spin-polarization vector, in particular for Rashba systems like GeTe. Together with a realistic model for the surface barrier potential, the one-step calculations were decisive to substantiate the $Ge_{1-x}Mn_xTe$ spectral features on both qualitative and quantitative levels. Ground state, as well as one-step calculations of substitutionally disordered $Ge_{1-x}Mn_xTe$ has been described by means of coherent potential approximation alloy theory. For simplicity the ground state calculations in Fig. 3b were based on the GeTe lattice structure[12] with added Mn substitutional doping on Ge sites, being the primary reason for the difference in the strength of the Rashba splitting α between theory and experiment reported in Fig. 3.

**Data availability.** The data that support the findings of this study are available on request from the corresponding author.

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

## Acknowledgements

This work was supported by the Swiss National Science Foundation Project No. PP00P2_1447421 and n.200021 146890. Financial support from the German funding agency DFG (SPP1666) and the German ministry BMBF (05K13WMA) is also gratefully acknowledged (H.V., H.E., J.B. and J.M.) G.S. and V.V.V. acknowledge support from the Austrian Science Funds (SFB-025, IRON). Financial support from the priority programme SPP 1666 of the Deutsche Forschungsgemeinschaft (Grant No. EB154/26-1) is gratefully acknowledged (H.V., H.E., J.B. and J.M.). J.M. acknowledges CENTEM PLUS (LO1402) project. F.B. and P.W. acknowledge the European Community's Seventh Framework Programme (FP7/2007–2013) under the grant agreement n.290605 (PSI-FELLOW/COFUND). P.W. acknowledges the technical support from N. Bingham and A. K. Suszka.

## Author contributions

V.N.S., J.H.D. and G.S. initiated and coordinated the project on equal level; H.V. performed the main calculations under supervision of J.M.; supporting calculations were carried out by J.B. and H.E.; SARPES experiments: J.K., S.M., M.F., A.P.W. and J.H.D.; soft-X-ray ARPES experiments: J.K., S.M., F.B. and V.N.S; PFM measurements: N.P.; SQUID measurements P.W.; sample growth and structural characterization: V.V.V. and G.S.; data analysis: J.K.; writing of manuscript: J.K., G.S., V.N.S and J.H.D. All authors extensively discussed the results and the manuscript.

## Additional information

**Competing financial interests:** The authors declare no competing financial interests.

