## [Peer review file · Nature Communications]

Reviewers' comments:

Reviewer #1 (Remarks to the Author):

The coexistence and coupling among various degrees of freedom provide fascinating physical properties and open routes for realizing multifunctional electronic devices. In this manuscript, the authors systematically research Ge_{1-x}Mn_xTe from the view point of multiferroic Rashba semiconductor. A sizable Zeeman gap around the Dirac point of the Rashba bands deriving from the collinear alignment of FE and FM polarization is proved by both the ARPES measurement and first-principles calculations. More importantly, the spin texture of the bands can be manipulated by magnetic fields and even electric ones, indicating its potential application in nonvolatile information storage.

This manuscript is well written, with great consistency among results from theoretical model, first-principles calculations and experiments, and novel for the research of both diluted ferromagnetic semiconductors and ferroelectric Rashba semiconductors. However, I have some suggestions for improvement, as listed below.

a) According to the Fig. 3, with the enhancement of Mn-doping, the Zeeman gap increases, accompanying with the decrease of the Rashba splitting. The microscopic origin has been well explained in the manuscript. However, when $x\text{Mn} > 10\%$, both the Zeeman gap and Rashba coupling constant tend to be saturated. What's the reason? Especially, the Zeeman gap gained from first-principles calculations still increases linearly. I don't think "the neglect of the strong band non-parabolicity" or "the GeTe lattice structure with added Mn substitutional doping on Ge-sites" would cause the difference. I recommend the authors to specify it.

b) Since that the magnetic-field-induced ferroelectric polarization reversal in Ge_{1-x}Mn_xTe has been reported in 2014 (cited as reference [8]), it is not surprisingly to realize the Pz-spin reorientation around Gamma point. I'm more interested in whether it is possible to experimentally manipulate it using electric field.

c) As a minor comment, I suggest that some errors should be corrected such as "First principles calculations" rather than "First principle calculations".

Overall, the paper is suitable for publication in Nature Communications.

Reviewer #2 (Remarks to the Author):

Krepasky and coworkers report on the multiferroic compound GeMnTe and the Rashba spin splitting in this system. This is interesting and practically speaking this is a nice study but there are few things that need to be considered before this work can be recommended for publication.

Most important is the magnetic control (Fig. 4d). In the motivation the magnetic switching of the out of plane and also in plane spin polarization is argued (Fig. 1g and 1h). But in the data only a small out of plane component is shown in the switching methodology. Authors should show this also for the in plane component to confirm their picture. I have some further questions about the spin polarization analysis. Why is the out of plane component so small? In Fig. 1c it seems practically not to be there. This must be justified to have the confidence in the claimed results. Why are large polarization found at 0 and -0.3 eV? They are larger than the claimed signal of the two bands. What values are found theoretically? These should be shown and compared while at the present no scale is given for the theory. In Fig. 1c what is the difference between the two sets of data that are included? Are these different samples? If it is just a different step size in the measurements it is misleading to include here as it implies different data supporting same

conclusion. Why is the out of plane polarization magnitude much larger for positive than negative applied magnetic fields?

The fits to Fig. 3 are not good enough to make any real conclusion. The authors should not claim to extract simply a Rashba parameter when the bands do not fit to the data. They claim this poor fitting is to be due to the non-parabolicity. Can they include this in the model? They must improve this fitting or remove it and not claim the parameters. But there also seem to be other inconsistencies. In the GeTe data (most obvious), additional bands and variations in the spectral weight would seem to be present. In their previous works these authors have discussed a lot about surface states in GeTe. Do these contribute to the states seen here?

I have some more minor comments.

The authors should cite the existing experimental work on GeTe (ref. 25) when they first introduce Rashba splitting in this compound as well as their own study (ref. 13).

Some of the claims that are made in the paper (first observation of Zeeman gap at Dirac point, excitations as Majorana fermions etc.) are very - lets say - over-sold and should be removed.

Reply to the Reviewers' comments:

Reviewer #1 (Remarks to the Author):

Comment 1: "The coexistence and coupling among various degrees of freedom provide fascinating physical properties and open routes for realizing multifunctional electronic devices. In this manuscript, the authors systematically research Ge_{1-x}Mn_xTe from the view point of multiferroic Rashba semiconductor. A sizable Zeeman gap around the Dirac point of the Rashba bands deriving from the collinear alignment of FE and FM polarization is proved by both the ARPES measurement and first-principles calculations. More importantly, the spin texture of the bands can be manipulated by magnetic fields and even electric ones, indicating its potential application in nonvolatile information storage.

This manuscript is well written, with great consistency among results from theoretical model, first-principles calculations and experiments, and novel for the research of both diluted ferromagnetic semiconductors and ferroelectric Rashba semiconductors. However, I have some suggestions for improvement, as listed below."

Answer:

We thank the referee for his or her appreciation of our results and for stressing the general importance of the field of study. As indicated below, we followed the suggestions of the referee to improve the manuscript.

Comment 2: "According to the Fig. 3, with the enhancement of Mn-doping, the Zeeman gap increases, accompanying with the decrease of the Rashba splitting. The microscopic origin has been well explained in the manuscript. However, when $x_{Mn} > 10\%$, both the Zeeman gap and Rashba coupling constant tend to be saturated. What's the reason? Especially, the Zeeman gap gained from first-principles calculations still increases linearly. I don't think "the neglect of the strong band non-parabolicity" or "the GeTe lattice structure with added Mn substitutional doping on Ge-sites" would cause the difference. I recommend the authors to specify it."

Answer:

Because of the length limitation, we dropped in the manuscript the following sentence from an earlier version of the manuscript that explained this saturation, which we now put back into the revised manuscript. The sentence now reads as:

...We also note that since the Zeeman gap appears to saturate for x_{Mn} around 10%, our conjecture is that higher Mn-doping might lead to Mn-phase segregation in the host GeTe lattice (17, 18) and that at higher Mn concentrations antiferromagnetic coupling between neighboring Mn atoms (x,y) reduces the average ferromagnetic moment per Mn atom.

17. Lechner, R. T. et al. Phase separation and exchange biasing in the ferromagnetic IV-VI semiconductor Ge_{1-x}Mn_xTe. *Applied Physics Letters* 97, – (2010).

18. Sato, K., Fukushima, T. & Katayama-Yoshida, H. Ferromagnetism and spinodal decomposition in dilute magnetic nitride semiconductors. *Journal of Physics: Condensed Matter* 19, 365212 (2007).

x. Y. Liu, S. K. Bose, and J. Kudrnovský, *Journal of Applied Physics* 112, 053902 (2012);

y. T Fukushima, H Shinya, H Fujii, K Sato, H Katayama-Yoshida and P H Dederichs, *J. Phys.: Condens. Matter* 27 (2015) 015501.

Comment 3: “Since that the magnetic-field-induced ferroelectric polarization reversal in $\text{Ge}_{1-x}\text{Mn}_x\text{Te}$ has been reported in 2014 (cited as reference [8]), it is not surprisingly to realize the P_z -spin reorientation around Gamma point. I'm more interested in whether it is possible to experimentally manipulate it using electric field.”

Answer:

We thank the referee for pointing out this issue. The gate control on (GeMn)Te samples grown on BaF_2 substrates is impossible because the substrate is a perfect insulator and thus provides no reference electrode. For this reason the films must be grown on a different, conductive, substrate. Furthermore, in-situ electric field switching requires a transparent top gate for SARPES experiments. We are working intensively to overcome these problems and expect to present conclusive results in the future. At this moment it surpasses technical capabilities and goes beyond the scope of the manuscript.

Comment 4: As a minor comment, I suggest that some errors should be corrected such as "First principles calculations" rather than "First principle calculations". Overall, the paper is suitable for publication in Nature Communications.

Answer:

We thank the referee for pointing out the typo. We have carefully checked the manuscript for similar mistakes and corrected them accordingly.

Reviewer #2 (Remarks to the Author):

Reviewer: “Krepasky and coworkers report on the multiferroic compound GeMnTe and the Rashba spin splitting in this system. This is interesting and practically speaking this is a nice study but there are few things that need to be considered before this work can be recommended for publication.”

Answer:

We thank the referee for his or her appreciation of our results. All his or her critique and concerns are addressed below.

Reviewer: “Most important is the magnetic control (Fig. 4d). In the motivation the magnetic switching of the out of plane and also in plane spin polarization is argued (Fig. 1g and 1h). But in the data only a small out of plane component is shown in the switching methodology. Authors should show this also for the in plane component to confirm their picture.”

Answer:

There are two reasons why in the B-field switching methodology we concentrated only on P_z . First, due to the complex GeTe surface electronic structure we demonstrate the spin-switching in the simplest case for normal emission because in this case the surface effects are optimally screened in photoemission. Second, the experimental setup at the used Cassiopee beamline at SOLEIL can measure only out-of-plane and radial spin components of the Rashba-type spin texture. We have no access to a SARPES experimental station which would allow us to study tangential spin-texture while changing the B-field.

However, thanks to the peculiar canted spin texture in (GeMn)Te as shown in the Supplementary Material, we verified the spin reorientation in the radial spin components. Figure RFig.1a depicted below shows SARPES EDCs for $\text{Ge}_{0.87}\text{Mn}_{0.13}\text{Te}$ obtained at 0.1 \AA^{-1} while switching the B-field ± 700 Gauss. In order to show how the spin-resolved EDCs relate to the band structure, RFig.1b shows an ARPES band map measured at $h\nu=22 \text{ eV}$ from uncapped samples. RFig.1c shows ARPES band maps measured at $h\nu=480$

eV from uncapped and RFig.1d $h\nu=840$ eV from capped film, with the EDCs indicated in green dashed line. We observe four spectral features: A is at E_F , B is above the Dirac point, C and D are below the Dirac point. Spectral features A and C belong to surface-resonance replica of the bulk GeTe states as sketched in RFig.1e, whereas B and D are bulk states. The surface resonance A at E_F does not change upon B-field switching. On the other hand spectral features B,C,D do indicate a change in the radial component.

RFig1: (a) $Ge_{0.87}Mn_{13}Te$ B_z -field control of the radial spin-polarization P_x for $k = 0.1 \text{ \AA}^{-1}$. (b) Second derivative ARPES band map measured at $h\nu=22$ eV from uncapped samples. (d) ARPES band maps measured at $h\nu=480$ eV from uncapped and $h\nu=840$ eV from capped film, with EDCs indicated in green color. (c) anatomy of the surface-resonance and bulk bands.

Due to the experimental energy resolution (80-90 meV), the P_x spin reorientation in B is less evident because of the dominant contribution of A which do not switch under the considered B-fields. Yet the trend indicated by purple (B+) and yellow (B-) arrows in P_x spin reorientation in RFig.1e is evident in our SARPES data in RFig.1a. However, these observations only indirectly suggest that the tangential spin-component, indicated by black arrows in RFig.1e, also behaves in the same way. For this reason, and also in order to keep our manuscript as simple as possible, we decided to include these results only in the Supplemental Material, also briefly explaining the intricate mixture of states with both bulk-derived and surface character.

Reviewer: "I have some further questions about the spin polarization analysis. Why is the out of plane component so small? In Fig. 1c it seems practically not to be there. This must be justified to have the confidence in the claimed results. Why are large polarization found at 0 and -0.3 eV? They are larger

than the claimed signal of the two bands. What values are found theoretically? These should be shown and compared while at the present no scale is given for the theory.”

Answer:

We assume that the reviewer had in mind data in Fig. 4c and its inset, not Fig. 1c. The theoretical P_z colorbar maximum and minimum values in Fig.4c are based on ground state calculations, and are a factor ten higher than measured. Our understanding is that this is due to inelastic scattering effects, the background, and the relatively bad energy and angular resolution of the SARPES experiment.

We would like to emphasize that our data in Fig.4c of the main text unambiguously show the P_z spin-polarization “wiggle” across the Zeeman gap upon B-field switching. It is outside the error margins and confirms the spin reorientation predicted by theory. In the revised manuscript we mention the reduction in spin polarization due to inelastic scattering.

Reviewer: “In Fig. 1c what is the difference between the two sets of data that are included? Are these different samples? If it is just a different step size in the measurements it is misleading to include here as it implies different data supporting same conclusion.”

Answer:

As already mentioned, the experimental P_z visualized via spin resolved energy distribution curve is manifested by the characteristic spin-polarization wiggle resolved across the Zeeman gap. All data in Fig.4c are measured from the same sample, which is $Ge_{0.87}Mn_{13}Te$ grown on BaF_2 . As indicated in the legend of Fig.4c, the difference is the step size. The smaller step size data is a verification of the larger step size data in a separate measurement. This is now clarified in the resubmitted manuscript.

Reviewer: “Why is the out of plane polarization magnitude much larger for positive than negative applied magnetic fields?”

Answer:

As seen in Fig.1f of main text, the $Ge_{0.87}Mn_{13}Te/BaF_2$ coercivity is about 400 Gauss. The maximum B-field accessible at the Cassiopee experimental station is 700 Gauss. Together with the remanent magnetic field of the experimental station we cannot guarantee that the effective magnetic field M inside the sample was the same when simply switching the B-field between ± 700 Gauss. This could be one reason for the difference. The other one, which is equally relevant, is that in the overall 3D spin dynamics of a multiferroic Rashba system coupled to external B and E-field, where the ferroelectric polarization P and magnetization M are antiparallel (not been found before in other systems), switching the M coupled to P might induce excitation affecting the M such that it comes back to equilibrium in different ways, possibly resulting in different spin-polarization magnitudes.

We have added a sentence to the manuscript to hint at such possibility.

Reviewer: “The fits to Fig. 3 are not good enough to make any real conclusion. The authors should not claim to extract simply a Rashba parameter when the bands do not fit to the data. They claim this poor fitting is to be due to the non-parabolicity. Can they include this in the model? They must improve this fitting or remove it and not claim the parameters. But there also seem to be other inconsistencies.”

Answer:

We appreciate the reviewer for critical comments on the fits based on the free-electron effective mass approximation. We chose the simplest possible model to reduce the number of fitting parameters. These type of fits are applicable only for small k-values and can thus be used to extract a Rashba parameter to

compare to literature, and to extract the Zeeman gap around $k=0$. We would like to point out that the typical Rashba parameter is only valid for parabolic bands. This is the reason why in the literature (Refs. 12 and 32) also for GeTe the Rashba parameter is only approximated for small k -values.

Ref.12 Di Sante, D., Barone, P., Bertacco, R. & Picozzi, S. Electric Control of the Giant Rashba Effect in Bulk GeTe. *Advanced Materials* 25, 509–513 (2013).

Ref.32 Picozzi, S. Ferroelectric Rashba Semiconductors as a novel class of multifunctional materials. *Frontiers in Physics* 2 (2014).

In order to explore the effect of the non-parabolic nature of the bands we altered the heavy Dirac fermion model typically used for IV-VI semiconductors to include a Rashba-like term and a Zeeman gap. The details of this phenomenological model are now explained in the Supplemental Material with resulting parameters, the fits are shown in the main text.

As expected, this model better reproduces the non-parabolic band shape, but only marginally affects the characteristic trend between the Rashba strength and Zeeman gap, as seen in the additional Figure SF2 in the Supplementary Material. We emphasize that for both the experiment and multiple scattering theory the Rashba strength parameter α_r for pure GeTe is found to be around 4.3 eV/\AA , the highest of so-far known materials, and in excellent agreement with theoretical prediction by S. Picozzi (*Frontiers in Physics* 2 (2014)). This reinforces that the fitting is correctly reflecting the intrinsic Rashba splitting in GeTe(111).

Reviewer: “In the GeTe data (most obvious), additional bands and variations in the spectral weight would seem to be present. In their previous works these authors have discussed a lot about surface states in GeTe. Do these contribute to the states seen here?”

Answer:

As mentioned above, the relevance the surface effects, which are particularly complex for the GeTe(111) surface, are reconsidered in more details by comparing photoemission data from capped and uncapped surfaces. Consistently with the existing GeTe photoemission data by Liebmann et.al in Ref. 25, figure RFig.1d shows that the uncapped surface unleashes otherwise suppressed surface effects on top of a bulk-like electronic structure as sketched in RFig. 1e. In other words, the soft-X ARPES from capped GeTe(111) surfaces allowed us to disentangle pure surface states from surface resonances and pure bulk states. Until more efficient spin detection schemes become available we are limited in spin-detection to the UV energy range where experiments from capped samples are difficult (or impossible) due to limited photoelectron escape depth. Based on the obtained experience, as also elucidated in Ref 13, we focus on regions, such as around $k=0$, where the influence of surface states is minimal.

Reviewer: “I have some more minor comments. The authors should cite the existing experimental work on GeTe (ref. 25) when they first introduce Rashba splitting in this compound as well as their own study (ref. 13). Some of the claims that are made in the paper (first observation of Zeeman gap at Dirac point, excitations as Majorana fermions etc.) are very - lets say - over-sold and should be removed.”

Answer:

The order of the references is now corrected.

As for the over-sold statements we are confident that our statement:

“To the best of our knowledge this is the first experimental confirmation of the opening of a Zeeman gap at the Dirac point in a system with strong ferromagnetic order, which was so far elusive in

magnetically doped topological insulators due to the lack of measurements justifying the ferromagnetic ordering of the dilute dopants at the ARPES measurement conditions”

is still relevant. However, we respect the reviewer’s opinion and altered this statement to focus on bulk materials where we are sure that this is the first direct observation of a gap opening at the Dirac point by ARPES. As for the Majorana fermions, we take the liberty to keep this statement in view of a recent publication by Wei Chen and Andreas P Schnyder, Majorana Edge States in Superconductor/Noncollinear Magnet Interfaces, arXiv:1504.02322, also dealing with Majorana fermions in multiferroics. We agree with the referee that any claims in this direction might currently appear premature, but we expect to encourage the scientific community to engineer superconducting heterostructures based on multiferroic (GeMn)Te.

REVIEWERS' COMMENTS:

Reviewer #1 (Remarks to the Author):

The authors have addressed my concerns properly. I therefore recommend its publication in NC.

Reviewer #2 (Remarks to the Author):

The points from my report has been nicely addressed and I am happy to suggest the paper should be published in Nature Communications. The only point on which I am not still convinced is the fitting of the data in Figure 3. It would be good if the authors could add in an error bar or statement a comment on how alpha may changes due to even the non-parabolic model to not agree well with the data.